# A genome-wide CRISPR screen identifies N-acetylglucosamine-1-phosphate transferase as a potential antiviral target for Ebola virus

Mike Flint [1], Payel Chatterjee[1], David L. Lin[2], Laura K. McMullan [1], Punya Shrivastava-Ranjan[1], Éric Bergeron [1], Michael K. Lo [1], Stephen R. Welch[1], Stuart T. Nichol[1], Andrew W. Tai [2,3] & Christina F. Spiropoulou[1]

There are no approved therapies for Ebola virus infection. Here, to find potential therapeutic targets, we perform a screen for genes essential for Ebola virus (EBOV) infection. We identify *GNPTAB*, which encodes the α and β subunits of N-acetylglucosamine-1-phosphate transferase. We show that EBOV infection of a *GNPTAB* knockout cell line is impaired, and that this is reversed by reconstituting GNPTAB expression. Fibroblasts from patients with mucolipidosis II, a disorder associated with mutations in *GNPTAB*, are refractory to EBOV, whereas cells from their healthy parents support infection. Impaired infection correlates with loss of the expression of cathepsin B, known to be essential for EBOV entry. GNPTAB activity is dependent upon proteolytic cleavage by the SKI-1/S1P protease. Inhibiting this protease with the small-molecule PF-429242 blocks EBOV entry and infection. Disruption of GNPTAB function may represent a strategy for a host-targeted therapy for EBOV.

[1] Viral Special Pathogens Branch, Division of High-Consequence Pathogens and Pathology, National Center for Emerging and Zoonotic Infectious Diseases, Centers for Disease Control and Prevention, 1600 Clifton Road, MS G-14, Atlanta, GA 30329, USA. [2] Department of Microbiology & Immunology, University of Michigan Medical School, Ann Arbor, MI 48109, USA. [3] Department of Internal Medicine, University of Michigan Medical School, Ann Arbor, MI 48109, USA. Correspondence and requests for materials should be addressed to M.F. (email: vfa3@cdc.gov) or to C.F.S. (email: ccs8@cdc.gov)

Ebolaviruses, such as Ebola virus (EBOV) and Reston virus (RESTV), and marburgviruses, such as Marburg virus (MARV) have single-stranded, negative-sense RNA genomes and are classified in the family *Filoviridae*. They cause sporadic outbreaks of severe hemorrhagic fever, with case fatality rates of up to 90%[1]. Although experimental vaccines and therapeutics appear promising[2–6], none are currently approved to protect from or treat filovirus infection.

EBOV particles are pleomorphic, but often have a long, filamentous structure, with a lipid envelope decorated with the viral glycoprotein (GP). In the current model of EBOV entry[7–11], particles bind to cells, are internalized by macropinocytosis and delivered to late endosomes/lysosomes (LE/LY). Here, cathepsins, cellular cysteine proteases, remove the glycan cap and mucin-like domains from GP. In the cleaved form, $GP_{CL}$, the receptor-binding site is exposed and interacts with the intracellular receptor Niemann-Pick C1 (NPC1). This interaction is thought to prime GP prior to fusion between the viral and cellular membranes, which results in the viral genome being deposited inside the target cell cytoplasm.

NPC1 is an integral LE/LY membrane protein that regulates the release of cholesterol from lysosomes. Loss of NPC1 function is associated with Niemann-Pick type C disease (NP-C), a rare autosomal recessive lysosomal storage disease (LSD) in which cells accumulate cholesterol and glycosphingolipids in their LE/LY. NPC1 is essential for filovirus infection as $NPC1^-$ patient fibroblasts were refractory to EBOV and MARV infection and heterozygous $NPC1^{+/-}$ mice survived a normally lethal challenge with mouse-adapted EBOV or MARV[12,13].

NPC1 was identified as a cellular factor required for EBOV entry both through a small-molecule inhibitor screen[14] and a loss-of-function gene-trapping method in a haploid cell line[12]. In the latter study, Carette and colleagues used cells transduced with a gene-trapping retrovirus and infected them with recombinant vesicular stomatitis virus (VSV) bearing the EBOV glycoprotein. The surviving cells were expanded and their insertion sites mapped. *NPC1* was the strongest hit in this screen, but other hits included genes from the homotypic fusion and vacuole protein sorting (HOPS) complex, which is involved in the fusion of endosomes to lysosomes; cathepsin B (*CTSB*)[15]; the lipid kinase *PIKFYVE*[16]; and *N*-acetylglucosamine-1-phosphate transferase alpha and beta subunits (*GNPTAB*). Another genome-wide screen used short-interfering RNA (siRNA) in A549 cells, using lentivirus bearing the MARV glycoprotein[17]. These authors reported *NPC1, VPS16* (a HOPS complex subunit), and cathepsin L (*CTSL*) as hits. A third screen used a short hairpin RNA (shRNA) method in 293T cells with infectious EBOV, and identified *NPC1, CTSB* and HOPS complex subunits. In addition to genes important for endosome maturation (*FIG4, PIKFYVE*), the lysosomal protein *BRI3*, and *RAB39B*, a GTPase involved in the regulation of vesicle trafficking were reported[18]. Finally, Martin and colleagues recently screened a siRNA library against an EBOV minigenome, a version of the genome with viral genes replaced by a reporter which is replicated by co-expressed viral proteins, to identify host factors important for replication and transcription in HEK293 cells[19]. This screen identified carbamoyl-phosphate synthetase 2, aspartate transcarbamylase and dihydroorotase (*CAD*) and the de novo pyrimidine synthesis pathway as being essential for EBOV replication and transcription.

Lysosomes mediate the degradation of extra- and intracellular material through an array of hydrolases, including proteases such as cathepsins[20]. Cathepsin B (CatB) is an essential factor for EBOV entry in cell culture, while cathepsin L (CatL) plays an accessory role. Together they mediate the stepwise cleavage of GP to $GP_{CL}$[15]. Many proteins are transported to lysosomes via the

mannose-6-phosphate pathway[21,22]. *N*-acetylglucosamine-1-phosphate transferase (GlcNAc-phosphotransferase) is a Golgi-resident enzyme that participates in the formation of mannose 6-phosphate on the glycans of newly synthesized proteins, which are then recognized by mannose-6-phosphate receptors and transported to lysosomes. GlcNAc-phosphotransferase is a 540 kiloDalton (kDa) heterotrimer with an $\alpha_2\beta_2\gamma_2$ structure; the *GNPTAB* gene encodes the α- and β-subunits, whereas the γ-subunit is encoded by *GNPTG*. To be activated, the GNPTAB polypeptide must be cleaved into the α and β subunits by the cellular protease SKI-1/S1P[23,24]. Low levels of GlcNAc-phosphotransferase activity are associated with the LSD mucolipidosis[25]. In this slowly progressive metabolic disease, lysosomal hydrolases are improperly targeted and, rather than being trafficked to lysosomes, become secreted. Mucolipidosis type II (MLII), also called inclusion-cell (I-cell) disease, is associated with variants in *GNPTAB* that result in little or no functional GlcNAc-phosphotransferase[26]. Mucolipidosis type III alpha/beta (MLIII alpha/beta), also called pseudo-Hurler polydystrophy, is associated with *GNPTAB* variants that retain some residual GlcNAc-phosphotransferase activity. MLIII alpha/beta progresses more slowly and is generally less severe than MLII[27]. A third LSD, mucolipidosis type III gamma (MLIII gamma), is caused by variants in *GNPTG*[28].

To identify further cellular factors required for EBOV infection that might be targets for host-directed therapies, we perform a whole-genome clustered regularly interspaced short palindromic repeat (CRISPR) screen, but rather than using recombinant VSV encoding EBOV GP, we use authentic infectious EBOV in a biosafety level 4 (BSL-4) laboratory. Our screen identifies GNPTAB as a host factor for EBOV infection. This requirement is confirmed in primary cells from MLII and MLIII patients with defective *GNPTAB* variants. Furthermore, we find that an inhibitor of the SKI-1/S1P protease required for GNPTAB activity blocks EBOV infection, suggesting that targeting GNPTAB may be a strategy for a host-targeted antiviral therapy for EBOV.

## Results

**A CRISPR screen for genes important for EBOV replication.** Genome-wide screens using gene-trapping, siRNA or shRNA methods have been used to identify host-factors required for filovirus infection[12,17–19]. Here, we performed a whole-genome CRISPR screen using the GeCKOv2 library in Huh7.5.1 cells[29], and used authentic EBOV (Mayinga strain) to infect the library-transduced cells at a multiplicity of infection (MOI) of 0.3 (Fig. 1). Extensive cell death occurred in the infected culture. Surviving cells were then expanded, their genomic DNA was extracted, and their single guide RNA (sgRNA) sequences were determined. Several genes with significantly enriched sgRNAs were identified (Table 1, Supplementary Data 1 & 2). *NPC1* was the top-ranked hit with 5 of 6 sgRNAs enriched. The second hit was Spinster-like 1 (*SPNS1*), which encodes a putative sugar transporter involved in lysosome function[30]. The third hit was Solute Carrier Family 30 Member 1 (*SLC30A1*), which encodes Zinc Transporter Protein 1[31]. Further work will be required to confirm a role for these proteins in EBOV infection. Other hits included the previously reported subunits of the HOPS complex *VPS16, VPS33A* and *VPS18* (Tables 1 and 2). *UVRAG* (Ultraviolet Radiation Resistance-Associated), which is required for influenza A and VSV entry[32] but not previously implicated in EBOV infection, was also found. *CTSL* was only a minor hit, while *CTSB* was not significant.

Altogether, these results suggested that our screen was capable of identifying genes required for EBOV infection. With three of six sgRNAs enriched, *GNPTAB* was a strong hit in our screen and

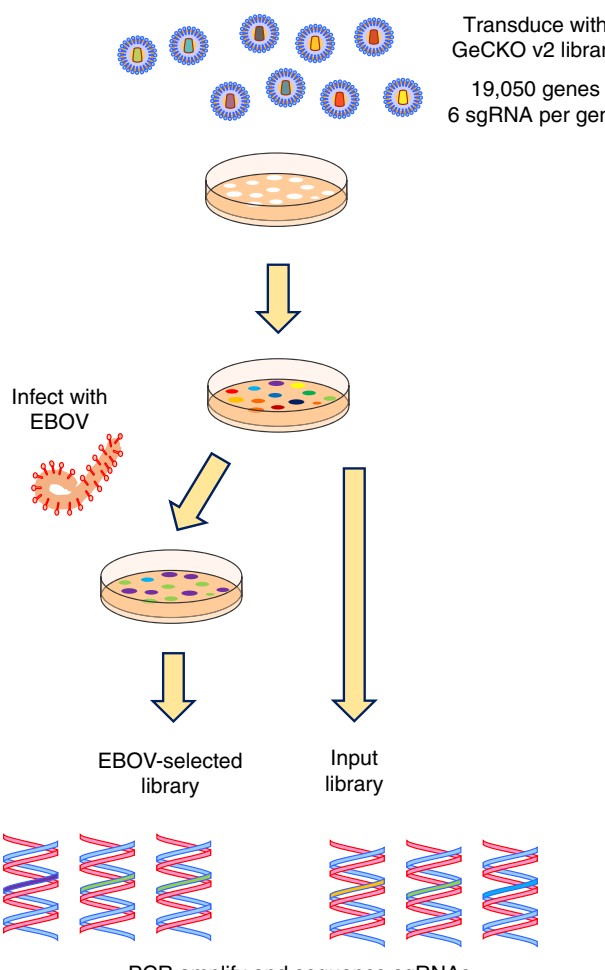

**Fig. 1** A CRISPR screen to select for knockout cells resistant to EBOV infection. The GeCKO v2 lentiviral CRISPR library was used to transduce Huh7.5.1 cells. The selected cells were infected with authentic EBOV and surviving cell colonies expanded. The sgRNAs were amplified and sequenced, and compared to those from the original transduced cells

also in that of Carette and colleagues[12], and we decided to further investigate its role in EBOV infection.

**GNPTAB is required for efficient EBOV infection.** To determine if GNPTAB has a role in EBOV infection, CRISPR genome editing was used to generate a clonal *GNPTAB*-knockout HAP1 cell line or, as a control, a *NPC1*-knockout cell line. Mutations in the gene of interest were confirmed by sequencing. Since the selection in our screen was based on the survival of transduced cells following EBOV infection, we first tested the viability of these cells after infection. The parental HAP1 cells, and GNPTAB⁻ and NPC1⁻ derivatives, were infected and their viability was determined 6 days post infection. Due to EBOV-induced cytopathic effect, the viability of the parental HAP1 cells was reduced to ~25% of that of mock-infected cells, whereas the GNPTAB⁻ and NPC1⁻ remained ~90% viable after infection (Fig. 2a). When these cells were infected with recombinant EBOV expressing the fluorescent reporter protein ZsGreen[33] (EBOV-ZsG), robust fluorescence was observed in parental HAP1 cells; fluorescence was lower in GNPTAB⁻ cells, and lowest in NPC1⁻ cells (Fig. 2b). This effect was not due to differences in cell growth, as mock-infected cultures had similar growth rates (Supplementary Figure 1a). We tested the ability of these cells to support infection by wild-type EBOV and found ~170-fold less virus yield in GNPTAB⁻ cells than in parental HAP1 cells (Fig. 2c). To determine the specificity of this effect, GNPTAB⁻ cells were infected with a panel of hemorrhagic fever viruses. These cells did not yield less reporter arenavirus Lassa virus (LASV-ZsG; Supplementary Figure 2a & b), flavivirus Alkhurma hemorrhagic fever virus (AHFV; Supplementary Figure 2d) or reporter phenuivirus Rift Valley fever virus expressing Green Fluorescent Protein (RVFV-GFP; Supplementary Figure 2e). The GNPTAB⁻ cells did, however, produce less Marburg virus (MARV-ZsG; up to 42-fold less than parental cells, Supplementary Figure 2f), RESTV (up to 207-fold less; Supplementary Figure 2g) and the paramyxovirus Nipah virus (NiV-ZsG, up to 640-fold less, Supplementary Figure 2h).

To confirm a role for GNPTAB in EBOV infection, lentiviruses expressing GNPTAB with a C-terminal myc-tag (GNPTAB-myc), or β-glucuronidase (GUS) as a control, were used to transduce parental HAP1 or GNPTAB⁻ cells. We did not detect

**Table 1 Selected hits from the CRISPR-Cas9 screen for genes important for authentic EBOV infection**

| Gene identifier | *P*-value | No. of enriched sgRNAs | Name | Comment |
|---|---|---|---|---|
| *NPC1* | 2.83E−07 | 5 | NPC Intracellular Cholesterol Transporter 1 | EBOV receptor |
| *SPNS1* | 2.83E−07 | 4 | Sphingolipid Transporter 1 (Putative). | Putative lysosomal transporter protein |
| *SLC30A1* | 1.42E−06 | 4 | Solute Carrier Family 30 Member 1 | Zinc transporter protein 1 |
| *VPS16* | 3.12E−04 | 3 | Vacuolar Protein Sorting 16 Homolog | HOPS complex subunit |
| *KLHDC3* | 1.94E−04 | 3 | Kelch Domain Containing 3 | |
| *STARD13* | 1.94E−04 | 3 | StAR Related Lipid Transfer (START) Domain Containing 13 | GTPase-activating protein |
| *GNPTAB* | 7.17E−04 | 3 | N-Acetylglucosamine-1-Phosphate Transferase Subunits Alpha and Beta | Catalyzes synthesis of mannose 6-phosphate on newly synthesized lysosomal enzymes |
| *ZNF646* | 1.14E−03 | 2 | Zinc Finger Protein 646 | |
| *VPS33A* | 9.60E−04 | 2 | Vacuolar Protein Sorting 33 Homolog A | HOPS complex subunit |
| *UVRAG* | 1.15E−03 | 2 | UV radiation resistance associated. | Required for Influenza A and VSV entry[32] |
| *VPS18* | 9.01E−03 | 2 | Vacuolar Protein Sorting 18 Homolog | HOPS complex subunit |
| *CTSL* | 5.82E−03 | 1 | Cathepsin L | |
| *CTSB* | 5.95E−02 | 1 | Cathepsin B | |

The gene identifier, *P* value, and number of enriched sgRNAs as calculated by MAGeCK analysis are indicated. The full MAGeCK output is provided in Supplementary Data Files 1 and 2

**Table 2 Comparison of the CRISPR-Cas9 screen with previously reported screens**

| | Carette[12] | Cheng[17] | Filone[18] | Martin[19] | Flint |
|---|---|---|---|---|---|
| Method | Gene-trap | siRNA | shRNA | siRNA | CRISPR |
| Cell line | HAP1 (human near-haploid chronic myelogenous leukemia) | A549 (human lung carcinoma) | 293T (human embryonic kidney with SV40 T-antigen) | HEK-293 (human embryonic kidney) | Huh7.5.1 (human hepatocyte carcinoma) |
| Challenge | VSV-EBOV-GP | HIV-MARV-GP | EBOV | EBOV minigenome | EBOV |
| NPC1 | ✓ | ✓ | ✓ | x[a] | ✓ |
| GNPTAB | ✓ | x | x | x[a] | ✓ |
| SPNS1 | x | x | x | x | ✓ |
| SLC30A1 | x | x | x | x | ✓ |
| UVRAG | x | x | x | x | ✓ |
| VPS11 | ✓ | x | x | x[a] | x |
| VPS16 | ✓ | ✓ | ✓ | x[a] | ✓ |
| VPS18 | ✓ | x | x | x[a] | ✓ |
| VPS33A | ✓ | x | x | x[a] | ✓ |
| VPS39 | ✓ | x | ✓ | x[a] | x |
| VPS41 | ✓ | x | ✓ | x[a] | ✓ |
| PIKFYVE | ✓ | x | ✓ | x[a] | x |
| BRI3 | ✓ | x | ✓ | x[a] | x |
| FIG4 | ✓ | x | ✓ | x[a] | x |
| BLOC1S1 | ✓ | x | x | x[a] | x |
| BLOC1S2 | ✓ | x | x | x[a] | x |
| SETDB1 | ✓ | x | x | x | x |
| MAGEF1 | ✓ | x | x | x | x |
| ARHGAP23 | ✓ | x | x | x | x |
| DDX39B | x | x | x | ✓ | x |
| CTSB | ✓ | x | ✓ | x[a] | x |
| CTSL | x | ✓ | x | x[a] | ✓ |
| EXT1 | x | ✓ | x | x | x |
| RAB39B | x | x | ✓ | x | x |
| CAD | x | x | x | ✓ | x |

A positive check for CRISPR-Cas9 screen is indicated if the *P*-value calculated by MAGeCK was <0.05. The Cheng and Filone screens did not report their full results, so only genes discussed by those authors are shown here
SV40 simian virus 40
[a]Expected, as minigenome does not recapitulate the viral entry mechanism

endogenous GNPTAB (three different commercially available antibodies were tested), but could detect GNPTAB-myc in the appropriately transduced cell populations (Fig. 3a, b). Each of the transduced cell lines still expressed NPC1 (Fig. 3b). When infected with EBOV, the GNPTAB⁻ cells transduced to express GUS (GNPTAB⁻ + GUS) remained refractory to EBOV-induced cell death, whereas the viability of those reconstituted with GNPTAB-myc was significantly reduced (Fig. 3c). In addition, the GNPTAB-myc reconstituted cells supported infection by EBOV-ZsG, whereas infection remained impaired in cells transduced to express GUS (Fig. 3d). The ability to support EBOV infection was not correlated with growth rate, as each grew similarly (Supplementary Figure 1b). In addition, the growth of LASV-ZsG was similar in each of the reconstituted cell lines (Supplementary Figure 2c). Altogether, our data indicate that GNPTAB is required for efficient EBOV infection of the haploid cell line HAP1.

**GNPTAB-deficient patient cells are refractory to infection.** Variants of *GNPTAB* are associated with the LSD mucolipidosis. We obtained primary fibroblasts from the families of three patients with MLII. For each family, cells from the healthy mother and father as well as the proband MLII patient were tested. The characteristics of these cells have been described in detail previously[34] and are summarized in Table 3.

Fibroblasts from three apparently healthy patients (GM00038, GM05659, and GM05565) and from three patients with NP-C (GM17914, GM03123, and GM18411) served as negative and positive controls, respectively. Each of the healthy control

fibroblasts supported infection by EBOV-ZsG (Fig. 4a–d, Supplementary Figure 3). In contrast, fibroblasts from 2 of the NP-C patients (GM17914 and GM18411) failed to support detectable EBOV replication (Fig. 4a, c). The third NP-C cell line (GM03123) supported a delayed, low level of infection (Fig. 4b). This patient was compound heterozygous at the *NPC1* allele, having the P237S and I1061T variants. While I1061T has been well characterized as associated with NP-C, the P237S variant was found to be non-pathogenic[35], consistent with the reduced level of infection observed. When fibroblasts from families of MLII patients were infected, in each case, the maternal and paternal cells supported EBOV-ZsG infection, but the proband's cells did not (Fig. 4a–c). The level of infection in cells from each of the three MLII probands was comparable to that in cells from NP-C patients. The growth rate of the fibroblast cells did not correlate with their ability to support EBOV infection (Supplementary Figure 5). Fibroblasts from two MLII probands (GM01586 and GM03066) and three healthy control patients were infected with LASV-ZsG, AHFV or RVFV-GFP. The growth of AHFV and RVFV-GFP in the MLII proband cells was unimpaired, being similar to that in the healthy fibroblasts (Supplementary Figure 6c & e). For LASV-ZsG, virus yield was similar in MLII and healthy fibroblasts for most of the time-course, but was modestly lower from the MLII cells at the day 4 time-point (Supplementary Figure 6a). Overall, this suggests that the impaired growth of EBOV in these cells was not simply due to a generalized lack of support for viral infection.

To further understand the relationship between GNPTAB dysfunction and EBOV infection, we tested fibroblasts from six

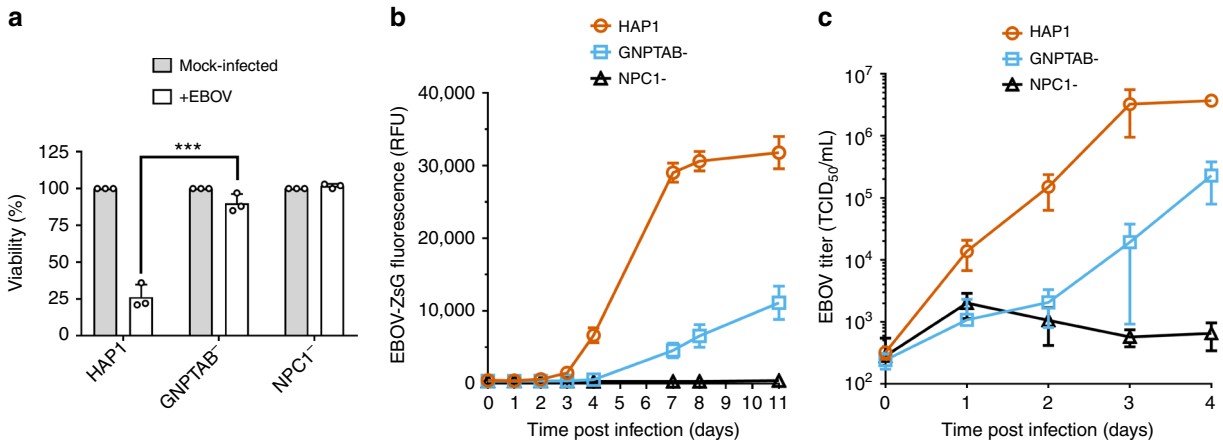

**Fig. 2** GNPTAB is required for efficient EBOV infection. **a** GNPTAB⁻ cells survive EBOV infection. Parental HAP1, GNPTAB⁻, or NPC1⁻ knockout cells were infected with wild-type EBOV at an MOI of 0.3, and viability relative to mock-infected cells was determined 6 days p.i. Data represent the mean ± s.d. of three independent experiments. Statistical analysis was performed with a two-tailed Student's t-test, with significance shown as ***, $P = 0.0005$. **b** Infection with the EBOV-ZsG reporter virus is reduced in GNPTAB⁻ knockout cells. Parental HAP1 (orange circles), GNPTAB⁻ knockout (blue squares) and NPC1⁻ knockout (black triangles) cells were infected with EBOV-ZsG at an MOI of 0.1 and ZsG fluorescence was measured over time. Data represent the mean ± s.d. of 4 biological replicates. A representative of three independent experiments is shown. **c** Wild-type EBOV growth is impaired in GNPTAB⁻ knockout cells. Parental and knockout HAP1 cells were infected with wild-type EBOV at an MOI of 0.5. Samples of culture supernatant were taken at intervals and the EBOV titer was determined. Data represent the mean ± s.d. of three biological replicates. A representative of two independent experiments is shown

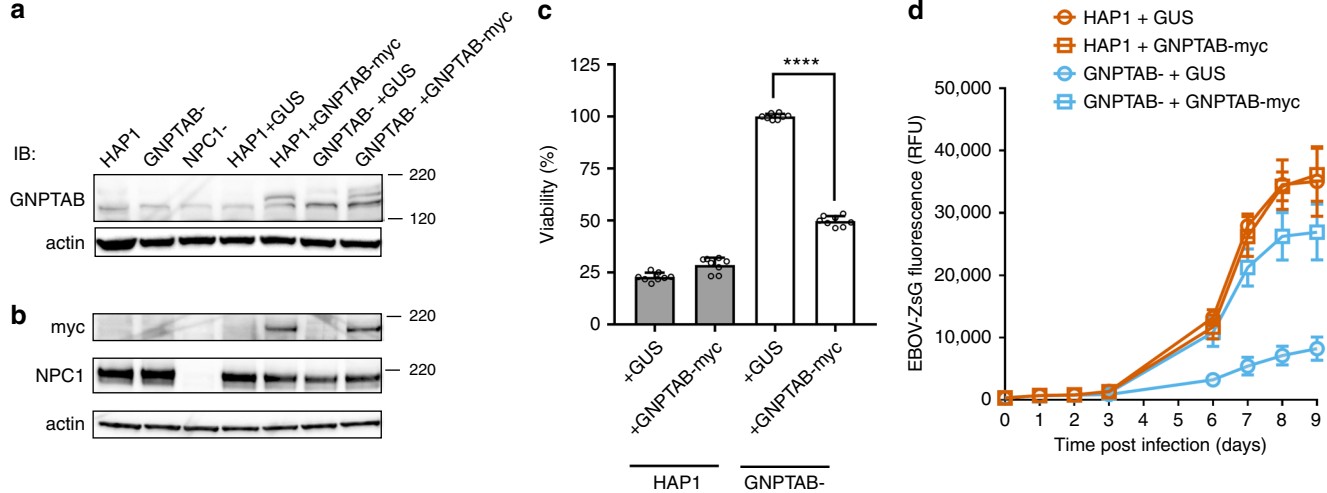

**Fig. 3** Reconstitution of GNPTAB in knockout cells restores EBOV infection. Parental HAP1 and GNPTAB⁻ knockout cells were transduced with lentiviruses to express β-glucuronidase (GUS) as a negative control, or myc-tagged GNPTAB (GNPTAB-myc). **a, b** Immunoblotting of lysates from HAP1, knockout and reconstituted cell lines. The migration of molecular mass markers in kDa is shown on the right. **c** Reconstitution of GNPTAB restores EBOV-induced cytopathic effect. Cells were infected and viability determined 6 days p.i. as for Fig. 2a. Data represent the mean ± s.d. of eight biological replicates. A representative of two independent experiments is shown. Statistical analysis was performed with a two-tailed Student's t-test with significance shown as **** $P < 0.0001$. **d** Reconstitution of GNPTAB restores EBOV infection. HAP1 and GNPTAB⁻ cells reconstituted to express GUS or GNPTAB-myc were infected with EBOV-ZsG virus and fluorescence was determined over time. Data represent the mean ± s.d. of six biological replicates. A representative of two independent experiments is shown

patients with MLIII, a less severe form of mucolipidosis, usually characterized by a residual level of GlcNAc-phosphotransferase activity. Cells from 5 MLIII alpha/beta patients with variants in *GNPTAB*, and from 1 MLIII gamma patient with a variant of *GNPTG*, were tested (Table 3). The cells from the MLIII patients supported EBOV-ZsG infection to varying degrees (Fig. 4d, Supplementary Figure 4); however, the level of infection for all was significantly lower than the least permissive healthy control we tested, GM00038. This effect was specific for EBOV, as neither LASV-ZsG nor RVFV-GFP was impaired in MLIII proband cells (Supplementary Figure 6b & f). Yields of AHFV were slightly

reduced from one of the three tested MLIII proband cells (GM02065), but the growth of AHFV in the other two was indistinguishable from that in the healthy control fibroblasts (Supplementary Figure 6d).

Overall, these data with primary fibroblast cells from both MLII and MLIII patients are consistent with a requirement for GNPTAB in EBOV infection of human cells.

**GNPTAB knockout correlates with loss of cathepsin activity.** GlcNAc-phosphotransferase is required for the correct

**Table 3 Primary human fibroblast cells used in this study**

| Individual | Classification | Family | Relation to proband | Gender | Age at sampling[a] | GlcNAc-Phospho-transferase activity (% of normal) | *GNPTAB* Allele 1 | *GNPTAB* Allele 2 | Comment |
|---|---|---|---|---|---|---|---|---|---|
| GM03066 | MLII | 34 | Proband | F | 23 fwk | 1 | FS288X | FS546X | — |
| GM00080 | — | 34 | Mother | F | 16 y | 66 | WT | FS546X | — |
| GM00081 | — | 34 | Father | M | 23 y | 96 | FS288X | WT | — |
| GM01586 | MLII | 1908 | Proband | M | 5 wk | <0.1 | FS1172X | FS1172X | — |
| GM01589 | — | 1908 | Father | M | — | 94 | FS1172X | WT | — |
| GM01590 | — | 1908 | Mother | F | — | 78 | FS1172X | WT | — |
| GM03112 | MLII | 1909 | Proband | F | 21 fwk | — | FS737X | FS1172X | — |
| GM02046 | — | 1909 | Mother | F | 25 y | 41 | FS737X | WT | — |
| GM02047 | — | 1909 | Father | M | 26 y | 52 | WT | FS1172X | — |
| GM03391 | MLIII gamma | — | Proband | M | 15 y | — | — | — | *GNPTG* 445delG/ 445delG |
| GM01494 | MLIII alpha/beta | 1012 | Proband | F[b] | 16 y[b] | 14 | K4Q | FS1172X | — |
| GM02425 | MLIII alpha/beta | — | Proband | M | 15 y | 1 | *FS211X (type2) | FS1172X | — |
| GM02065 | MLIII alpha/beta | — | Proband | M | 9 y | 2 | FS745X | *FS1085X (type 2) | — |
| GM00113 | MLIII alpha/beta | — | Proband | F | 2 y | 12 | K4Q | K4Q | — |
| GM02558 | MLIII alpha/beta | 501 | Proband | F | 9 y | 3 | Q278X | *FS1085X (type 2) | — |
| GM17914 | Niemann-Pick disease, type C1 | — | Proband | F | 7 y | — | — | — | *NPC1* I1061T/ 424insGA |
| GM03123 | Niemann-Pick disease, type C1 | — | Proband | F | 9 y | — | — | — | *NPC1* P237S/ I1061T |
| GM18411 | Niemann-Pick disease, type C1 | — | Proband | F | — | — | — | — | *NPC1* W833X/ I1061T |
| GM05659 | Apparently healthy, non-fetal | — | n/a | M | 1 y | — | — | — | — |
| GM05565 | Apparently healthy, non-fetal | — | n/a | M | 3 y | — | — | — | — |
| GM00038 | Apparently healthy, non-fetal | — | n/a | F | 9 y | — | — | — | — |

Patient data are derived from Kudo and colleagues[34] and the Coriell Institute for Medical Research (https://www.coriell.org), and is reprised here for convenience
MLII mucolipidosis type II, MLIII mucolipidosis type III, — no data available, n/a not applicable
[a]Age at the time of tissue collection. In the case of fetal samples, an estimated gestational age is in fetal weeks (fwk) since conception
[b]Differs from reference (Kudo et al.), but confirmed by the Coriell Institute

intracellular localization of lysosomal proteins like cathepsins. To determine if GNPTAB's role in EBOV infection might be through cathepsin function, we used immunoblotting to detect CatB and CatL in lysates from parental, knockout, and reconstituted HAP1 cells. Bands consistent with the mature single-chain forms of CatB (30 kDa) and CatL (32 kDa) were not detected in GNPTAB⁻ cells, but expression of both was restored following reconstitution with GNPTAB-myc (Fig. 5a, Supplementary Figure 7a & b). Since CatB is essential for Zaire-EBOV entry, we performed assays for CatB function using fluorescent substrate peptides that become dequenched after proteolysis. We used inhibitors of CatB ($B_i$) and CatL ($L_i$), to demonstrate that CatB activity was specifically detected. Lysates from the parental HAP1 cells and the NPC1⁻ knockout cells possessed robust CatB activity (Fig. 5b) that was inhibited by adding $B_i$ but not $L_i$. In contrast, CatB activity was significantly lower in the GNPTAB⁻ knockout cells (Fig. 5b). CatB activity was restored by GNPTAB-myc expression (Fig. 5c). These data are consistent with GNPTAB influencing EBOV infection by modulating the activity of CatB, a critical factor for EBOV entry.

**An inhibitor of GNPTAB processing blocks EBOV infection.** The SKI-1/S1P protease is required for the cleavage of GNPTAB

to enable GlcNAc-phosphotransferase activity[23,24]. We hypothesized that PF-429242, an inhibitor of SKI-1/S1P[36], would inhibit EBOV infection by blocking GNPTAB activation. First, we determined if PF-429242 could prevent cleavage of the GNPTAB precursor (PT-αβ) to its alpha (PT-α) and beta subunits (PT-β). Cells transfected to express GNPTAB-myc and treated with increasing concentrations of PF-429242 exhibited less PT-αβ processing, as evidenced by a reduced amount of PT-β-myc (Fig. 6a). PF-429242 had no effect on NPC1 expression levels (Fig. 6b). We examined the effect of PF-429242 on EBOV-GP-mediated entry using pseudotyped lentiviral particles bearing the EBOV-GP (EBOVpp). The compound blocked EBOVpp entry with a mean 50% effective concentration ($EC_{50}$) ± s.d. of 0.80 ± 0.17 μM ($n = 3$; Fig. 6c), whereas the inhibition of particles pseudotyped with VSV-G protein (VSVpp; $EC_{50} = 12.80 ± 0.82$ μM, $n = 3$; Fig. 6c) was similar to the effect on cell viability (Fig. 6d). PF-429242 inhibited EBOV-ZsG virus infection with a mean $EC_{50}$ of 0.95 ± 0.57 μM ($n = 6$), with a selectivity index (SI) of 15 (Fig. 6d). A control compound, 2′-C-methylcytidine (2′-CMC)[37], blocked AHFV-induced CPE in A549 cells, while PF-429242 had no effect (Supplementary Figure 8a). Similarly, RVFV-GFP fluorescence in Huh7 cells was inhibited by the 2′-deoxyfluorocytidine (2′-dFC) control[38], but the inhibition

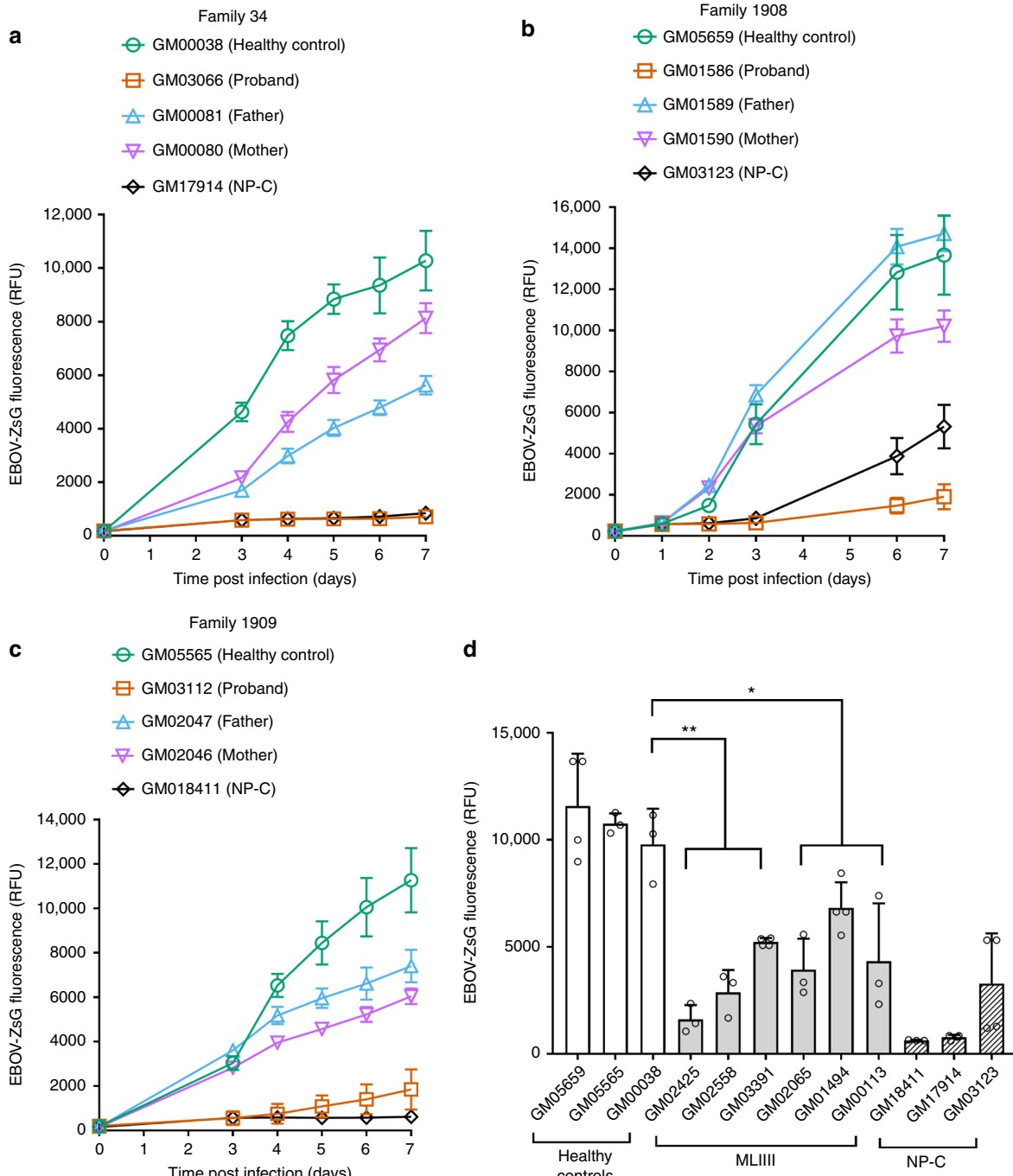

**Fig. 4** Fibroblasts from mucolipidosis patients do not efficiently support EBOV infection. Fibroblast cells were infected with the EBOV-ZsG reporter virus at an MOI of 1 and ZsG fluorescence was followed over time. **a** Family 34, **b** family 1908, **c** family 1909. Data represent the mean ± s.d. of eight biological replicates. A representative of at least two independent experiments is shown. **d** Fibroblasts from MLIII patients, compared to cells from healthy controls and NP-C patients. ZsG fluorescence was measured 7 days post infection. Data represent the mean ± s.d. from at least three independent experiments. Statistical analysis was performed with a two-tailed Student's $t$-test with significance shown as ** $P \leq 0.005$, * $P \leq 0.05$

induced by PF-429242 was consistent with cytotoxicity (compare Fig. 6d and Supplementary Figure 8b). Thus, the SKI-1 inhibitor PF-429242 blocked GNPTAB maturation and specifically inhibited EBOV-GP-mediated entry and EBOV infection.

## Discussion
Several genome-wide screens to identify host-factors required for filovirus infection have been reported (Table 2). We performed a

CRISPR-Cas9 screen to attempt to identify genes important for authentic EBOV infection. Genome-wide screens can frequently yield discordant results[39,40], with different techniques, cell lines and challenge methods likely being sources of discrepancy. For example, retroviral gene-trapping and CRISPR-Cas9 methods are capable of inducing gene knockouts, whereas si- and shRNA use RNA interference (RNAi) to degrade transcripts. For a protein with a long half-life, or a very abundant transcript, a knockout screening method may be more effective than a knockdown. This

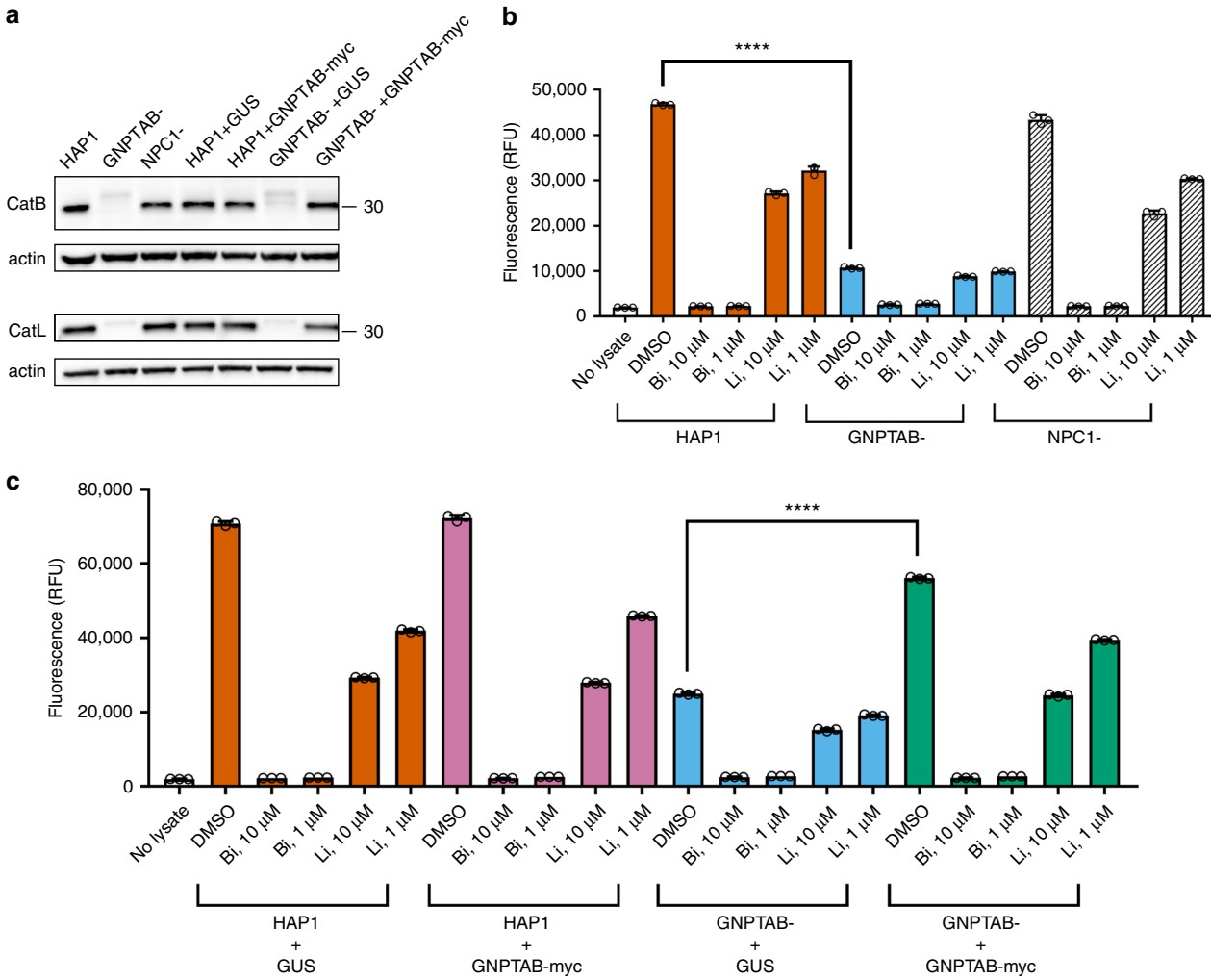

**Fig. 5** Cathepsins are reduced in GNPTAB⁻ knockout cells, but restored upon GNPTAB reconstitution. **a** Immunoblotting of lysates from parental, knockout and reconstituted cells for CatB and CatL. The migration of a 30 kDa molecular mass marker is shown to the right. **b** CatB activity in lysates from parental HAP1, or knockout GNPTAB- or NPC1-cells. Cells were treated with cathepsin B inhibitor (Bi), cathepsin L inhibitor (Li), at 10 or 1 μM, or DMSO vehicle control for 1 h at 37 °C, before lysis and incubation with fluorescent CatB peptide substrate. After 1 h incubation at room temperature, fluorescence was measured. Data represent the mean ± s.d. of three technical replicates. A representative of three independent experiments is shown. **c** CatB activity in lysates from reconstituted cells. Lysates from transduced HAP1 or GNPTAB⁻ cells were generated and tested for CatB activity, as for panel **b**. Data represent the mean ± s.d. of three technical replicates. A representative of two independent experiments is shown. For both panels **b** and **c**, statistical analysis was performed with a two-tailed Student's t-test with significance shown as **** $P \leq 0.0001$

may be why *GNPTAB* was a hit in two screens that used gene knockout, but not in those using RNAi (Table 2). We used authentic EBOV as the challenge virus. While this has benefits in recapitulating the genuine infection process, EBOV does have a propensity to form defective-interfering particles[41] which may cause false-positive hits. It should be noted that in addition to genes required for EBOV infection, genes required for the death of EBOV-infected cells should also be identified in our screen.

The *GNPTAB* gene encodes the α and β subunits of GlcNAc-phosphotransferase, an enzyme that participates in the formation of mannose-6-phosphate on proteins to direct their transport to lysosomes. We confirmed the role of *GNPTAB* in EBOV infection in both HAP1 cells and primary fibroblasts from individuals with mucolipidosis, a genetic disorder associated with mutations in *GNPTAB*. While GNPTAB is required for the correct localization of numerous lysosomal hydrolases, and we cannot exclude the involvement of other factors, lack of EBOV infection correlated with the loss of CatB activity, which is known to be required for

EBOV entry. A model for the role of GNPTAB in EBOV entry is shown in Fig. 7.

Lysosomal cysteine proteases are required to activate GP from all filoviruses, but different viruses use different cathepsins. For example, entry mediated by EBOV GP was strongly dependent upon CatB, whereas MARV GP was more dependent upon CatL[42]. RESTV GP was not sensitive to the loss of CatB and CatL, but was inhibited by the broad cysteine protease inhibitor E-64, suggesting that a different cysteine protease is required for RESTV infection[42]. We found that RESTV infection of GNPTAB⁻ cells was impaired (Supplementary Figure 2g), consistent with RESTV GP activation being dependent upon a lysosomal protease, transported there through the action of GNPTAB. Additional cysteine proteases are required even for viruses that use CatB, as viruses bearing precleaved glycoproteins remain sensitive to inhibition by E-64[42–45]. This is consistent with a multi-step model where initial cleavages by CatB and L expose the NPC1 - binding domain, but an additional E64-sensitive step(s), mediated

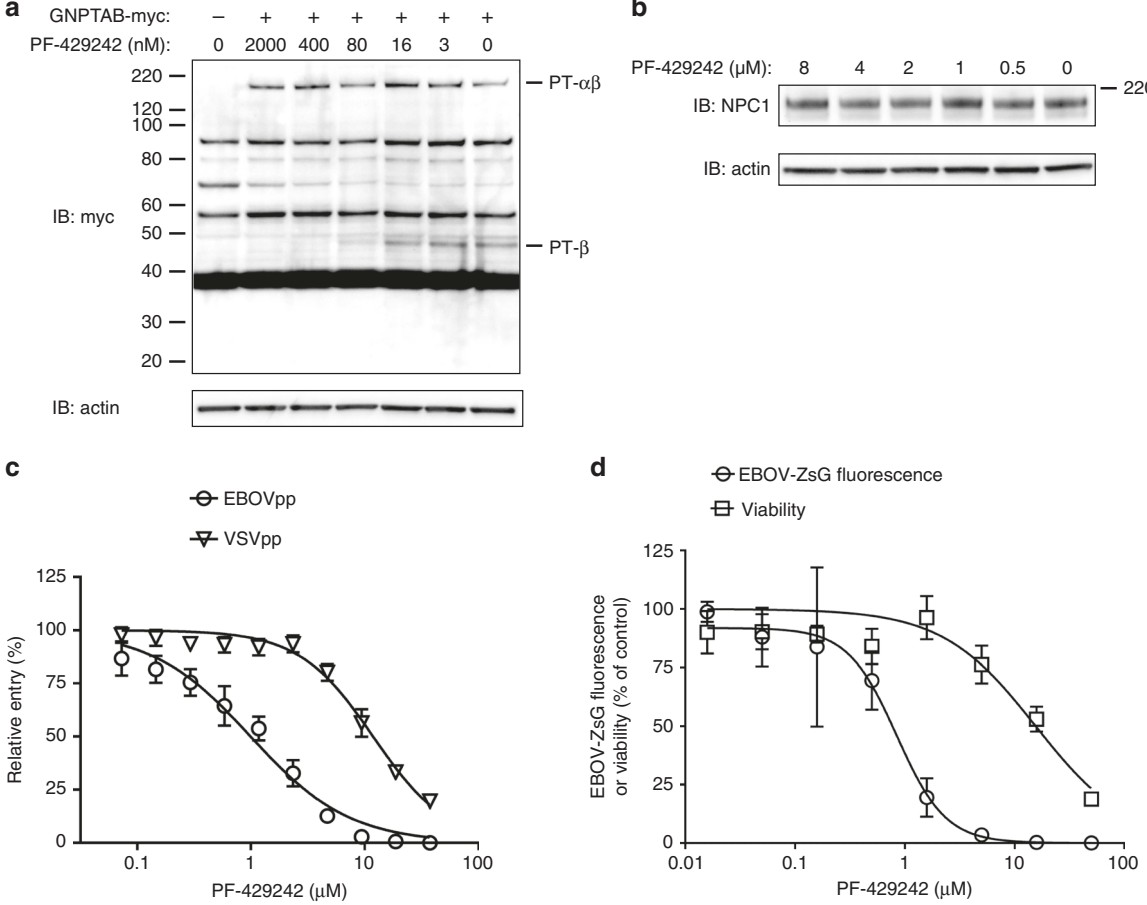

**Fig. 6** Blocking GNPTAB cleavage with a SKI-1/S1P inhibitor blocks EBOV infection. **a** PF-429242 blocks processing of GNPTAB. Huh7 cells were transfected with a plasmid expressing GNPTAB-myc. Twenty-four hours post transfection, PF-429242, or DMSO vehicle, was added to the indicated concentrations. Cell lysates were harvested 48 h post transfection and subjected to immunoblotting. **b** PF-429242 does not alter NPC1 expression levels. Huh7 cells were treated with PF-429242 at the indicated concentrations for 24 h before cell lysates were harvested and subjected to immunoblotting. **c** PF-429242 blocks EBOV-GP-mediated entry. Varying concentrations of PF-429242 were added to Huh7 cells 1 h prior to the addition of luciferase-encoding lentiviral particles pseudotyped with the Zaire-EBOV-GP (EBOVpp), or with VSV-G protein (VSVpp). Three days later luciferase activity was determined. Data represent the mean and s.d. of values relative to the DMSO vehicle control, from 8 biological replicates. A representative of 3 independent experiments is shown. **d** PF-429242 inhibits infection by EBOV-ZsG reporter virus. Huh7 cells were treated with varying concentrations of PF-429242, before being mock-infected or infected with EBOV-ZsG reporter virus. Three days post-infection, cell viability or ZsG fluorescence was measured, respectively. Data represent the mean and s.d. of values relative to the DMSO vehicle control, from four biological replicates. A representative of six independent experiments is shown

by another lysosomal protease, is required for fusion to occur. Interestingly, wild-type, $CTSB^{-/-}$, and $CTSL^{-/-}$ mice were equally susceptible to lethal challenge with mouse-adapted EBOV[46]. This suggests that in vivo, at least for mouse-adapted EBOV in mice, GP may be cleaved by proteases other than CatB and CatL. This could be due to differences in the specificities of the different species' cathepsins, or may also be the case during human infection.

One effect of modulating GNPTAB function may be the inactivation of multiple cathepsins, which, unlike a small molecule directed against a single cathepsin, might have the effect of blocking multiple filoviruses. We showed that GNPTAB-knockout reduced MARV infection, though not to the extent seen with EBOV (Supplementary Figure 2f) and also RESTV-infection (Supplementary Figure 2g). Inhibition of GNPTAB would also be predicted to block infection by other viruses that require cathepsins, such as Nipah virus[47,48] (Supplementary Figure 2h), SARS-coronavirus[49], and reoviruses[50]. Cathepsins are thought to be involved in antigen processing, generating peptides for presentation by major histocompatibility complex class II

molecules on the surface of antigen-presenting cells[51–53]. Consequently, suppressing GNPTAB function may prevent the initiation of an adaptive immune response, though we have been unable to find any reference to immune suppression in the context of mucolipidosis patients. Clearly, for an antiviral strategy targeting GNPTAB, as with any therapy, the consequences for patient physiology need to be fully evaluated.

We found that PF-429242, an inhibitor of the SKI-1/S1P protease required for GNPTAB maturation, inhibited EBOV-GP-mediated entry and EBOV infection. PF-429242 was originally developed to reduce cholesterol and fatty acid synthesis, as SKI-1/S1P is also responsible for processing sterol regulatory element-binding proteins, which are major transcriptional regulators of cholesterol synthesis. PF-429242 inhibited SKI-1/S1P activity with an $IC_{50}$ of 0.2 μM and cholesterol synthesis with an $EC_{50}$ of 0.5 μM[36]. We recently demonstrated that the cholesterol-lowering drugs statins can suppress EBOV infection by interfering with proper GP processing[54]. Similarly, the cholesterol-reducing function of PF-429242 may also prevent correct GP glycosylation, and this mechanism may contribute to its anti-EBOV

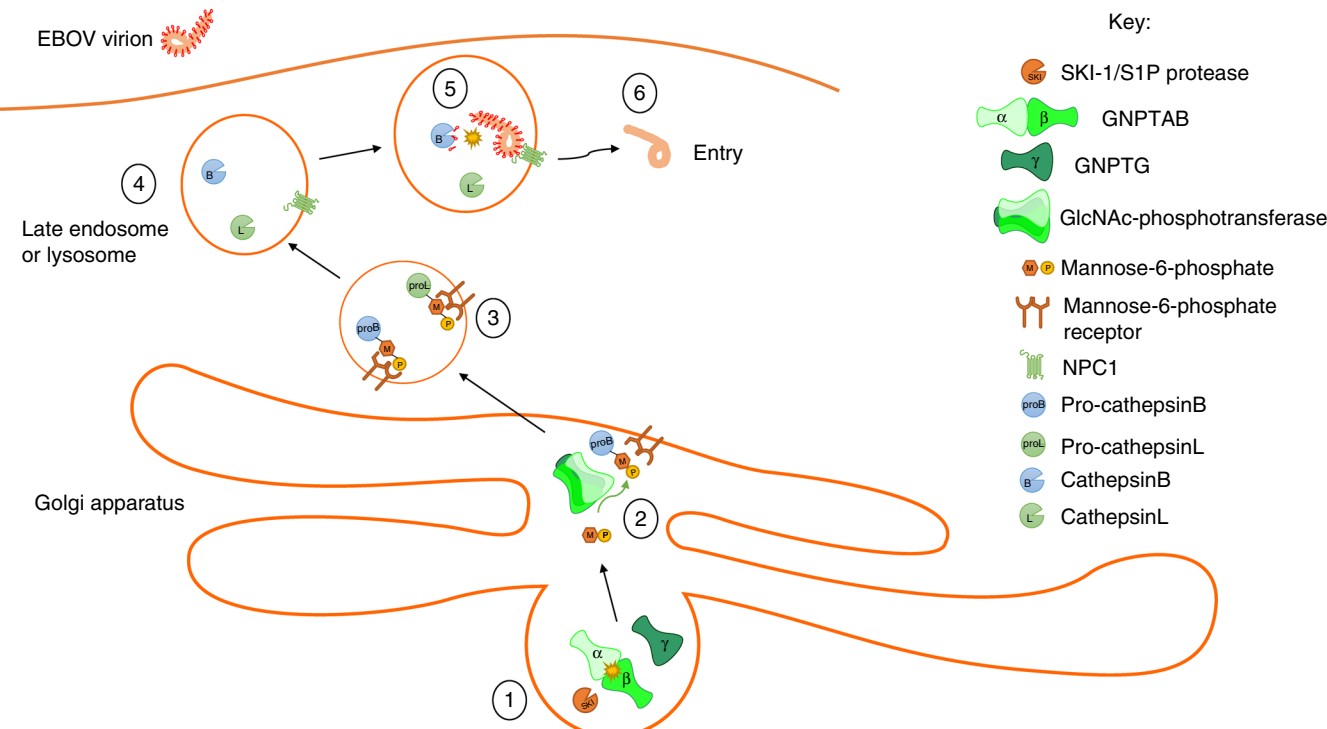

**Fig. 7** Schematic of a model for the role of GNPTAB in EBOV entry. (1) The SKI-1/S1P protease cleaves the GNPTAB precursor to yield the individual α and β subunits, which then assemble with the γ subunit to form the GlcNAc-phosphotransferase enzyme. (2) GlcNAc-phosphotransferase catalyzes the transfer of mannose-6-phosphate onto the glycans of proteins destined to be transported to the lysosome, such as pro-cathepsin B and pro-cathepsin L. (3) The mannose-6-phosphate receptor recognizes its substrates and mediates their transport to the lysosome. (4) Once in the lysosome, pro-cathepsins cleave to yield the mature cathepsins. (5) The activated cathepsins cleave the EBOV GP, removing the glycan-cap and mucin-like domains, and allowing the receptor-binding site to interact with NPC1, priming the GP for fusion. (6) Following fusion, the viral nucleocapsid enters into the cell cytoplasm where replication can be initiated. Blocking GlcNAc-phosphotransferase activity, either by inhibition of the SKI-1/S1P protease at (1), or by the absence of functional enzyme in (2), has the effect of preventing EBOV entry

activity. Indeed, PF-429242 inhibits infection by a number of viruses in cell culture because SKI-1/S1P is required for the maturation of their glycoproteins[55,56] or because they are dependent upon high levels of cellular cholesterol for entry[57]. The inactivation of GNPTAB and the improper trafficking of lysosomal proteins are additional mechanisms that may contribute to the antiviral activity of this compound. Notably, however, PF-429242 has a high clearance rate (75 mL/min/kg) and low oral availability (~5%)[36], and failed to protect mice from severe fever upon challenge with thrombocytopenia syndrome virus[58].

Our data suggest that GNPTAB could be the subject of a host-targeted anti-EBOV therapy. Using a compound like PF-429242 to block GNPTAB function, and consequently lysosomal cathepsin activities, should inhibit EBOV infection. Such treatment would essentially involve the deliberate induction of a LSD to prevent EBOV from entering via lysosomes, as has been suggested previously for suppression of NPC1. While GNPTAB deficiency is associated with mucolipidosis, the clinical manifestations occur much more slowly than the acute course of EBOV infection, and the risks of short-term GNPTAB inhibition may be outweighed by the benefits for a disease with a high case fatality rate. We found that fibroblasts from several patients with MLIII, the less severe form of mucolipidosis associated with residual levels of GlcNAc-phosphotransferase activity, did not efficiently support EBOV growth, suggesting that even a partial reduction in GNPTAB activity might be sufficient to reduce infection.

In conclusion, we have demonstrated that GNPTAB is required for efficient EBOV infection, suggesting a host pathway that may be targeted as part of an anti-EBOV therapeutic strategy.

## Methods

**Biosafety**. All work with infectious virus was conducted in a BSL-4 laboratory at the Centers for Disease Control and Prevention (CDC, Atlanta, USA). All laboratorians adhered to international practices appropriate for this biosafety level. Experiments involving cDNA encoding viral sequences were approved by the CDC Institutional Biosafety Committee.

**Cells, viruses, and reagents**. Huh7 cells were from Apath, LLC (Brooklyn, USA) and Huh7.5.1 cells were a kind gift from Francis Chisari (The Scripps Research Institute, USA), both were maintained in Dulbecco's Modified Eagle Medium (DMEM) supplemented with 10% (v/v) fetal calf serum (FCS; Hyclone, GE Healthcare, Marlborough, USA), non-essential amino acids and penicillin and streptomycin (ThermoFisher, Grand Island, USA). HAP1 cells, both the parental and the GNPTAB- and NPC1- derivatives, were obtained from Horizon Genomics (Cambridge, UK). HAP1 cells were maintained in Iscove's Modified Dulbecco's Medium (IMDM; ThermoFisher) supplemented with 10% (v/v) FCS and penicillin and streptomycin. PCR amplification and Sanger sequencing (at Eurofins Genomics, Louisville, USA) was used to confirm mutations in the genes of interest (primer sequences are listed in Supplementary Table 1). Human fibroblast cells were from the National Institute of General Medical Sciences (NIGMS) Human Genetic Cell Repository at the Coriell Institute for Medical Research (Camden, USA), and were cultured in Eagle's Minimal Essential Medium (EMEM; American Type Culture Collection, Manassas, USA) supplemented with 15% (v/v) FCS. A549 and Vero E6 cells were from the CDC core facility and maintained in DMEM supplemented with 10% (v/v) FCS. All cells were cultured at 37 °C in 5% $CO_2$ and were tested periodically to rule out mycoplasma contamination.

Wild-type EBOV, strain Mayinga, was launched from recombinant DNA by co-transfection of Huh7 cells with plasmids encoding the full-length genome and codon-optimized EBOV proteins NP, L, VP35, VP30 and T7 RNA polymerase[33,59]. Four days post transfection, the clarified medium for the transfected cells was passed onto fresh Huh7 monolayers. The medium from these cells was harvested 3 days later and titered on Huh7 cells. A stock was then generated following infection of Huh7 cells at an MOI of 0.005, to avoid the emergence of defective-interfering particles. The recombinant viruses EBOV-ZsG, LASV-ZsG, MARV-ZsG, NiV-ZsG, and RVFV-GFP, engineered with genes encoding fluorescent

reporter proteins in the genome plasmid, were similarly launched by following plasmid transfection[33,59–62]. PF-429242 was from Tocris Biosciences (Minneapolis, USA) or Cayman Chemicals (Ann Arbor, USA). Oligonucleotide primers were from Integrated DNA Technologies, Inc. (Skokie, USA).

**Pooled CRISPR screen.** The pooled CRISPR screen was performed twice at a library coverage of ~150× each time. The GeCKO v2 library was gift from Feng Zhang, acquired from Addgene (#1000000048; Watertown, USA). VSV-G-pseudotyped GeCKOv2 lentiviruses were generated at the University of Michigan vector core. Each packaged GeCKOv2 A or B lentiviral half-library was used to transduce 16 million Huh7.5.1 cells as a MOI of 0.3 in 10-cm dishes[29]. After transduction, the cells were selected for 6 days with 2 μg/mL puromycin, then frozen and transferred to CDC. Following revival of each half-library, $10^7$ transduced cells were infected with wild-type EBOV (Mayinga strain) at an MOI of 0.3. Three days post infection, the cells were passaged at a 1:5 ratio. Surviving cells were expanded, and lysed in TriPure (Roche Applied Science, Mannheim, Germany), to inactivate EBOV present in the sample. The lysates were then transferred from BSL-4 containment to a BSL-2 laboratory and genomic DNA (gDNA) was extracted using the TriPure method. The sgRNA sequences were amplified in two sequential PCR reactions, both using Herculase II enzyme (Agilent, Santa Clara, USA). For the first reaction, for gDNA from EBOV-selected cells and from input library cells, 13 replicate 100 μL reactions were set up, each with 10 μg of extracted gDNA, and products were amplified over 18 cycles using primers PCR#1-forward and –reverse (listed in Supplementary Table 1). The appropriate PCR#1 reactions were then pooled and used as template in the second reaction with forward (F01-06) and reverse (R01-R04) primers (Supplementary Table 1) to add Illumina adaptors and barcodes[63], over 15 cycles. The resulting PCR products were purified on an agarose gel (Zymoclean Gel DNA Recovery Kit, Zymo Research, Irvine, USA), analyzed using a Bioanalyzer High Sensitivity DNA Analysis Kit (Agilent) and quantified using a NEBNext Library Quant Kit for Illumina (New England Biolabs, Ipswich, USA). Products were mixed in equimolar concentrations and sequencing was performed on the Illumina HiSeq 2500 at Omega Bioservices (Norcross, USA). CLC Genomics Workbench 10 (Qiagen, Redwood City, USA) was used to trim adapter sequences and demultiplex FASTQ files. The 20 bp sgRNA sequences were analyzed using the MAGeCK software package v0.5.7[64]. A list of the non-targeting sgRNAs included in the library was used as the normalization method.

**Immunoblotting.** Medium was removed from cell monolayers, the cells were washed with PBS, then lysed in RIPA buffer (25 mM Tris pH 7.6, 150 mM NaCl, 1% Nonidet-P40, 1% sodium deoxycholate, 0.1% sodium dodecyl sulfate; ThermoFisher) supplemented with protease inhibitors (cOmplete, Roche). Insoluble material was removed by centrifugation at 18,000×$g$ for 20 min at 4 °C. Lysates were resolved on 4–12% Bis-Tris NuPAGE Novex gels (ThermoFisher), before being transferred to polyvinylidene fluoride membranes. Membranes were blocked with 5% nonfat dry milk in phosphate-buffered saline solution (PBS; Thermo-Fisher) supplemented with 0.1% Tween-20 (ThermoFisher). Antibodies used for immunoblotting were anti-GNPTAB (#PA5-69636, ThermoFisher at 1:1000), anti-myc (clone 9E10, #MA1-980, ThermoFisher at 1:1000), anti-NPC1 (#108921, Abcam, Cambridge, USA at 1:500), anti-actin, (clone AC-15, #A5441, Sigma-Aldrich, St. Louis, USA at 1:1000), anti-CatB (#31718, Cell Signaling Technology, Danvers, USA at 1:1000) and anti-CatL (#AF952, R&D Systems, Minneapolis, USA at 1:5000). Secondary antibodies were horseradish peroxidase (HRP)-conjugated anti-rabbit and HRP-conjugated anti-mouse (both 1:50,000, Jackson ImmunoResearch, West Grove, USA). Bound antibodies were detected using Super Signal West Dura enhanced chemiluminescence (ECL) substrate (ThermoFisher) and visualized using a ChemiDoc MP Imaging System and Image Lab v3.2.1 software (Bio-Rad). Uncropped scans of the immunoblots shown in Fig. 5 are supplied as Supplementary Figure 7.

**Lentiviral transductions.** The GNPTAB-myc open reading frame was amplified from plasmid pcDNA3.1-GNPTAB[65], a gift from Thomas Braulke (Addgene plasmid #78108), cloned into a transfer vector and from there into the pSCRPSY lentiviral plasmid (a kind gift from Charlie Rice, Rockefeller University). Lentiviruses were packaged by co-transfection with VSV-G expression plasmid in Lenti-X 293T cells (Takara Bio, Mountain View, USA). Parental and GNPTAB-knockout HAP1 cells were transduced at an MOI of 0.3, and selected with 1 μg/mL puromycin starting three days post-transduction.

**Virus assays.** For assays to measure EBOV-induced CPE, HAP1 cells were seeded at 5000 cells per well of a 96-well plate. The next day, cells were infected with wild-type EBOV (strain Mayinga) at an MOI of 0.3. Three days post-infection (p.i.), cells were trypsinized and passed at 1:10 ratio into an opaque white 96-well plate. Six days p.i. cell viability was determined using CellTiter-Glo according to the manufacturer's instructions (Promega, Madison, USA).

For time-courses using the fluorescent reporter viruses, EBOV-ZsG, LASV-ZsG, and RVFV-GFP, HAP1 cells were seeded at 5000 cells, or fibroblasts at 3000 cells,

per well of an opaque 96-well plate. The following day, HAP1 cells were infected with EBOV-ZsG virus at an MOI of 0.1 or fibroblasts were infected at an MOI of 1 (titer being determined using the healthy control fibroblasts GM05659). Fluorescence, representing the intensity within the infected cells, was measured over time using a Synergy H1MD plate reader (BioTek, Winooski, USA). For time-courses measuring viral titers, HAP1 cells were seeded and infected as for the fluorescent viruses, with samples of the culture medium quantified for virus titers by 50% tissue culture infectious dose ($TCID_{50}$) assays in Vero E6 cells by the method of Reed and Muench.

Assays testing for the inhibition of EBOV-GP-mediated entry were performed with pseudotyped HIV particles[66]. Particles were generated by co-transfection of Lenti-X 293T cells with Lipofectamine 2000 (ThermoFisher) and plasmids expressing EBOV-GP or VSV-G, and a HIV luciferase reporter vector containing defective Nef, Env and Vpr (obtained through the NIH AIDS Reagent Program, Division of AIDS, NIAID, NIH: pNL403.Luc.R⁻E⁻ from Dr Nathaniel Landau[67,68]), at 1:32 ratio. Transfected cell supernatants were harvested at 48 and 72 h post transfection, pooled, passed through at 0.45 μM polyethylsulfone filter to remove debris, and aliquots were frozen at −80 °C. The Lenti-X p24 Rapid Titer kit (Takara Bio) was used to determine the HIV core protein p24 content of stocks. To measure pseudotyped entry, Huh7 cells were seeded at 10,000 cells per well of a 96-well plate. The following day, the cells were treated with varying concentrations of PF-429242 for 1 h at 37 °C, then pseudotyped particles were added (6 ng of HIV p24 matrix protein per well). Six hours later, the inoculum was removed and replaced with complete medium. Firefly luciferase activity was determined using Bright-Glo luciferase assay system (Promega) 3 days later.

Assays for inhibition of EBOV-ZsG infection, were performed in Huh7 cells[69]. Cells were seeded at 3000 per well of a 384-well plate. The following day, the cells were treated with varying concentrations of PF-429242 for 2 h, before being infected with EBOV-ZsG virus at an MOI of 0.3. ZsGreen fluorescence was measured 3 days later using a Synergy H1MD plate reader.

**Assay for CatB activity.** In the assay to measure CatB activity[50], cells were detached from plates using enzyme-free cell-dissociation buffer (ThermoFisher) and $2.5 \times 10^6$ cells were transferred to 1.5 mL microfuge tubes in 1 mL of medium supplemented with CatB inhibitor ($B_i$; Z-Phe-Ala-CH$_2$F, Calbiochem #342000, MilliporeSigma, Burlington, USA), CatL inhibitor ($L_i$; Z-Phe-Tyr(t-Bu)-diazo-methylketone, Calbiochem #219427) or DMSO vehicle. Cells were incubated at 37 °C for 1 h, then washed twice with PBS and resuspended in lysis buffer (100 mM sodium acetate pH 5, 1 mM EDTA, 0.5% v/v Triton X-100). Insoluble material was pelleted in a microfuge at 18,000×$g$ for 20 min, and the supernatant harvested. In a well of a black, opaque 96-well plate, 20 μL of cell lysate was added to 80μL of reaction buffer (100 mM sodium acetate pH 5, 1 mM EDTA, 4 mM dithiothreitol) and 100 μL substrate solution (100 μM of CatB substrate peptide, Z-Arg-Arg-AMC, Calbiochem #219392 resuspended in lysis buffer). The reaction was incubated at room temperature for 1 h, and fluorescence was measured (excitation 380 nm, emission 460 nm) using a Synergy H1MD plate reader.

**Graphing and statistics.** For concentration-response plots, GraphPad Prism 7.0 (GraphPad Software, La Jolla, USA) was used to fit a 4-parameter equation to semi-log plots of the data and to derive the concentration of compound that inhibited 50% of the measured effect ($EC_{50}$). All statistical analyses were also performed in GraphPad Prism 7.0. For Student's $t$-test, a $P$ value of <0.05 was considered statistically significant. In bar graphs, the individual data points are overlaid.

**Disclaimer.** The findings and conclusions in this report are those of the authors and do not necessarily represent the official position of the Centers for Disease Control and Prevention.

## Data availability
All relevant data are available from the corresponding authors upon request.

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

## Acknowledgements

We thank César Albariño and Lisa Guerrero for providing the reporter Marburg virus and Tanya Klimova for assistance with editing this manuscript. This work was supported by National Institutes of Health grants RO1DK097374 (A.W.T.) and the Molecular Mechanisms of Microbial Pathogenesis Training Program 5T32AI007528 (D.L.L.).

## Author contributions

Conceptualization: M.F., A.W.T. and C.F.S.; Investigation: M.F., P.C., D.L.L., P.S.-R., E.B., M.K.L. and S.W.; Funding acquisition: S.T.N. and A.W.T.; Resources: L.K.M., P.S.-R., M.K.L. and S.W.; Writing of the original draft: M.F.; Review and editing: all authors.

## Additional information

**Competing interests:** The authors declare no competing interests.

