## [Peer Review File · Nature Communications]

Reviewers' comments:

Reviewer #1 (Remarks to the Author):

Flint et al conduct a CRISPR screen to identify host genes required for Ebola virus replication. They focus on investigating one of these genes, also identified in previous screens, GNPTAB. As it was known that GNPTAB participates in cathepsin trafficking to the lysosome, and that Ebola virus entry requires cathepsin activity in the lysosome for entry, the mechanistic connection between GNPTAB requirement and Ebola virus entry is perhaps self-evident. In any case, the authors clearly demonstrate a defect in Ebola virus replication in fibroblasts of patients with GNPTAB defects, caused by reduction of cathepsin in the lysosome, and demonstrate that inhibition of functional processing of the encoded protein by GNPTAB also inhibits Ebola virus replication. These are data of interest to general readers. Nevertheless, there are a few aspects that would make the study more informative (see below)

Major comments

1. The complete dataset of the CRISPR screen should be made available, and the authors need to show and discuss the overlapping between their CRISPR screen and the previous screens that have been conducted to identify host genes participating in Ebola virus replication.
2. In the first experiments using CRISPR cells, the authors use Lassa virus infection as a control virus infection not affected by loss of GNPTAB expression. Such irrelevant virus controls should also be used in experiments with cells from patients with genetic defects in GNPTAB and in experiments with inhibitors of SKI-1/S1P protease.

Minor comments

1. Abstract: "The gene GNPTAB, which encodes subunits of N-acetylglucosamine-1-phosphate transferase...". The specific subunits encoded by the gene should be mentioned in the abstract.
2. Lines 127-129: "When these cells were infected with recombinant EBOV expressing the fluorescent reporter protein ZsGreen27 (EBOV-ZsG), robust fluorescence was observed in parental HAP1 cells; fluorescence was lower in GNPTAB- cells, and lowest in NPC1- cells" Do the authors refer to number of green cells, to intensity of GFP within infected cells or to both?

Reviewer #2 (Remarks to the Author):

In their manuscript entitled "A genome-wide CRISPR screen identified N-acetylglucosamine-1-phosphate transferase as a potential antiviral target for Ebola virus" Flint et al use a genetic screen to identify host factors required for viral infection. Important benefit of the host factors survey described here compared to previous screens are:

1. the use authentic Ebola virus instead of a recombinant virus that mimics Ebola virus in certain molecular (but not morphological) aspects
2. The use of a CRISPR library that may target genes more efficiently than RNAi approaches and with more equal gene coverage than the gene-trap approach in haploid cells
3. the use of a relevant human cell type

The main hit that was identified is NPC1 which is the known entry receptor and the second most significant outlier is GNPTAB which the authors proceed to study. They show that GNPTAB is needed for infection and that patient cells resist infection. In addition, perturbation of GNPTAB function by targeting an activating protease also impairs infection. It is suggested that this effect is primarily caused by impaired cathepsin activity in the GNPTAB cells (although Cathepsin itself was not found as a hit in the screen and the activating protease S1P neither suggesting that the screen that was carried out was perhaps not very sensitive?). To prove that the phenotype of GNPTAB-deficient cells is caused by defective Cathepsin activity the authors could pre-cleave the virus with Cathepsins and test if this overcomes the infection barrier that is observed in GNPTAB-deficient cells.

In principle the experiments are solid and conform that GNPTAB is needed for the infection of authentic Ebola virus. The results are relevant but not highly surprising: as the authors mention, GNPTAB was already among the hits found in an earlier study using VSV-Ebola. Also, GNPTAB is known to be required for the function of Cathepsins which are well-know host factors required for Ebola virus infection. Therefore, this study mostly confirms a function for GNPTAB that was already expected. It would have been more interesting if this screen using authentic Ebola virus would have pointed out entirely new host factors that were needed for infection. However, due to the limited sensitivity of the screen we can also not conclude that VSV-Eb and authentic Ebola virus have the exact same host factor requirements for entry.

Together, I think that the experiments are solid and the conclusions are relevant and should be published. However, I am not convinced that the work should be published in Nature Communications.

Reviewer #3 (Remarks to the Author):

In the manuscript "A genome-wide CRISPR screen identifies N-acetylglucosamine-1-2 phosphate transferase as a potential antiviral target for Ebola virus" by Flint et al. the authors identify N-acetylglucosamine-1-2 phosphate transferase (GNPTAB) as an essential host factor for the Ebola virus (EBOV) life cycle, and show that inhibiting the processing of this factor results in inhibition of EBOV entry and infection. The manuscript is very well written, and the experiments are straightforward and adequately address the scientific questions the authors pose. From a scientific point of view, there is little to be criticized, with exception of a few points as listed below.

The major issue with this study with respect to publication in Nature Communications is that of impact, importance to a wider field, and novelty. As the authors mention themselves, GNPTAB has been identified in a previous genome-wide haploid genetic screen in human cells (Carette et al., 2011, PMID 21866103). Further (and not mentioned by the authors), there are a number of other genome-wide screens for filovirus host-factors, although this study is to the best of my knowledge the first using CRISPR technology. Filone et al. reported in 2015 a genome-wide shRNA-based screen using infectious EBOV, although they did not identify GNPTAB (Filone et al., 2015, PMID 25646658). Similarly, the entry of a close relative of EBOV, Marburg virus (MARV), was studied using a genome-wide siRNA screen together with MARV GP-pseudotype viruses (Cheng et al., 2015, PMID 26596270). Again, GNPTAB was not reported as an identified factor; however, the authors of that study did not disclose the complete data set, but just a chosen subset of identified factors. Finally, Martin et al. recently reported a genome-wide siRNA screen for host factors important for the life cycle of EBOV, although in their screen they only assessed a role in genome replication and transcription, and not in entry (Martin et al., 2018, PMID 30081931).

In addition, to me it is not clear whether the (re-)identification of GNPTAB will be of interest to a wider field of virologists outside of the filovirus community. Only very limited data is presented that this factor plays a role for other viruses, although the authors discuss this possibility. Also, as a target for treatment GNPTAB it is certainly interesting; however, based on the presented data it is not clear whether such an approach in the end will be feasible. Therefore, while the final decision whether this manuscript meets the criteria of publication in Nature Communication has to be made on the editorial level, I personally am doubtful that this is the case.

Individual points of criticism that should be addressed prior to publication are:

1) Data availability: Since the authors performed a genome-wide screen, they should provide all the results of this screen. It is hard to believe that they identified only 4 host factors in their screen! If this is indeed the case, they should discuss why their analysis yielded only so few hits, and disclose the primary data in order to allow others to mine this dataset.

2) line 52: The authors' use of nomenclature is wrong. This sentence should read: "Ebolaviruses, such as Ebola virus (EBOV) and Reston virus (RESTV), and marburgviruses, such as Marburg virus (MARV), have a single-stranded ..."

3) The authors mention that RESTV apparently does not use Cathepsin B or L for its entry. Is this virus then still dependent on GNPTAB? The authors could (and should) easily check this in their pseudotype assay.

4) The authors should mention other (genome-wide) screens for filovirus host factors, and discuss why these screens did not pick up GNPTAB.

We thank all the reviewers for their comments on our manuscript. In light of their suggestions, we have undertaken an extensive revision of our draft and believe that it is much improved as a result. We hope that we have addressed all of the concerns that were raised. Major changes include:

- Since the initial analysis of our data, the MAGeCK software used to determine hit significance has been upgraded several times. We have repeated the analysis of our screening results using the latest version, which uses a different normalization than previously. This has led to some changes in the ranking of hits identified by the software; for example, GNPTAB falls from 2nd on the list to 10th.
- As suggested reviewers 1 and 3, we now provide the full results of the CRISPR screen, both for the individual sgRNA sequences and for the genes they are designed to target. These are supplied as Supplementary Files 1 and 2, respectively.
- To put the results of our screen into the context of the work of others, we have substantially revised the text in multiple places and have included citations to the previously published papers mentioned by the reviewers. We have also included a new table comparing the hits from our own and previously published screens (new Table 2).
- We have included additional irrelevant virus controls in several experiments. These include time-courses of two additional hemorrhagic fever viruses the flavivirus Alkhurma hemorrhagic fever virus (AHFV; new Fig. S2d) and the phenuivirus Rift Valley fever virus (RVFV; new Fig. S2e), neither of which were affected by the knockout of GNPTAB in HAP1 cells. We tested primary fibroblast cells from 5 different mucopolidosis patients (2 with type II, 3 with type III) with 3 viruses unrelated to Ebola (new Fig. S6). Finally, we demonstrate that the SKI-1/S1P protease inhibitor PF-429242 does not inhibit infection by either AHFV or RVFV (new Fig. S8).
- To investigate the requirement for GNPTAB for a glycoprotein not dependent upon cathepsins B and L, we used Reston virus (RESTV) to infect GNPTAB-knockout cells (new Fig. S2g). We found that RESTV infection was impaired in these cells, consistent with a different lysosomal protease being important for RESTV glycoprotein activation.

Our specific responses to each of the reviewers' comments are below:

Reviewer #1 (Remarks to the Author):

Flint et al conduct a CRISPR screen to identify host genes required for Ebola virus replication. They focus on investigating one of these genes, also identified in previous screens, GNPTAB. As it was known that GNPTAB participates in cathepsin trafficking to the lysosome, and that Ebola virus entry requires cathepsin activity in the lysosome for entry, the mechanistic connection between GNPTAB requirement and Ebola virus entry is perhaps self-evident. In any case, the authors clearly demonstrate a defect in Ebola virus replication in fibroblasts of patients with GNPTAB defects, caused by reduction of cathepsin in the lysosome, and demonstrate that inhibition of functional processing of the encoded protein by GNPTAB also inhibits Ebola virus replication. These are data of interest to general readers. Nevertheless, there are a few aspects that would make the study more informative (see below)

Major comments

1. The complete dataset of the CRISPR screen should be made available, and the authors need to show and

discuss the overlapping between their CRISP screen and the previous screens that have been conducted to identify host genes participating in Ebola virus replication.

We have reanalyzed the results from the CRISPR screen and now provide the updated results both at the sgRNA level and at the gene level, in supplementary files 1 and 2, respectively. We now discuss the overlap between our own screen and those of others, listing hits and comparing them in our new Table 2. In addition, in our revised discussion, we speculate on some reasons why different genome-wide screens may yield different results.

2. In the first experiments using CRISPR cells, the authors use Lassa virus infection as a control virus infection not affected by loss of GNPTAB expression. Such irrelevant virus controls should also be used in experiments with cells from patients with genetic defects in GNPTAB and in experiments with inhibitors of SKI-1/S1P protease.

We have now performed several experiments with additional hemorrhagic fever viruses acting as irrelevant controls. In addition to the previously presented Lassa virus (LASV), we now show that the growth of the flavivirus Alkhurma hemorrhagic fever virus (AHFV) and the phenuivirus Rift Valley fever virus (RVFV) is unaffected in GNPTAB-knockout HAP1 cells (new Fig.S2d & e respectively). This expands the number of viruses we demonstrate to be unimpaired by GNPTAB-knockout to three (LASV, AHFV and RVFV), compared with three that are affected (Ebola viruses, Marburg and Nipah).

For cells from patients with genetic defects in GNPTAB, we used fibroblasts from 5 different mucopolipidosis patients (2 mucopolipidosis type II and 3 mucopolipidosis type III), and from 3 healthy controls, and performed infections with LASV, AHFV and RVFV in each (new Fig. S6). These data demonstrate that mucopolipidosis patient cells are competent to support the growth of multiple viruses, and that the impaired growth of EBOV that we observed in these cells was not simply due to a generalized lack of support for viral infection.

Finally, we used irrelevant control viruses in concentration-response experiments with the SKI-1/S1P protease inhibitor. Arenaviruses make use of the host SKI-1/S1P protease for cleavage of their glycoprotein, and PF-429242 inhibits LASV growth (Urata et al, 2011, PMID 21068251). So instead of LASV, we show that AHFV-induced cytopathic effect was inhibited by a control compound 2'-CMC, but not by PF-429242 (new Fig. S8a). Similarly, RVFV-GFP infection was blocked by a control compound, but the inhibition seen with PF-429242 was consistent with an effect on cell viability (compare Fig. 6d with new Fig. S8b).

Altogether, this new data with additional viruses helps demonstrate the specificity of the loss of GNPTAB function on Ebola virus infection.

Minor comments

1. Abstract: "The gene GNPTAB, which encodes subunits of N-acetylglucosamine-1-phosphate transferase...". The specific subunits encoded by the gen should be mentioned in the abstract.

The specific subunits encoded by *GNPTAB* are now indicated in the abstract: "The gene *GNPTAB*, which encodes the α and β subunits of N-acetylglucosamine-1-phosphate transferase...".

2. Lines 127-129: "When these cells were infected with recombinant EBOV expressing the fluorescent reporter protein ZsGreen27 (EBOV-ZsG), robust fluorescence was observed in parental HAP1 cells; fluorescence was lower in GNPTAB- cells, and lowest in NPC1- cells" Do the authors refer to number of green cells, to intensity of GFP within infected cells or to both?

For these experiments, we did not count green cells, but used a plate reader to measure the total ZsGreen fluorescent signal in wells of microtiter plates, so the signal represents the intensity of fluorescence within the infected cells. To clarify this, we have amended the appropriate methods section to read: "Fluorescence, representing the intensity within the infected cells, was measured over time using a Synergy H1MD plate reader (BioTek, Winooski, USA)."

Reviewer #2 (Remarks to the Author):

In their manuscript entitled "A genome-wide CRISPR screen identified N-acetylglucosamine-1- phosphate transferase as a potential antiviral target for Ebola virus" Flint et al use a genetic screen to identify host factors required for viral infection. Important benefit of the host factors survey described here compared to previous screens are:

- 1. the use authentic Ebola virus instead of a recombinant virus that mimics Ebola virus in certain molecular (but not morphological) aspects**
- 2. The use of a CRISPR library that may target genes more efficiently than RNAi approaches and with more equal gene coverage than the gene-trap approach in haploid cells**
- 3. the use of a relevant human cell type**

The main hit that was identified is NPC1 which is the known entry receptor and the second most significant outlier is GNPTAB which the authors proceed to study. They show that GNPTAB is needed for infection and that patient cells resist infection. In addition, perturbation of GNPTAB function by targeting an activating protease also impairs infection. It is suggested that this effect is primarily caused by impaired cathepsin activity in the GNPTAB cells (although Cathepsin itself was not found as a hit in the screen and the activating protease S1P neither suggesting that the screen that was carried out was perhaps not very sensitive?). To prove that the phenotype of GNPTAB-deficient cells is caused by defective Cathepsin activity the authors could pre-cleave the virus with Cathepsins and test if this overcomes the infection barrier that is observed in GNPTAB-deficient cells.

In principle the experiments are solid and conform that GNPTAB is needed for the infection of authentic Ebola virus. The results are relevant but not highly surprising: as the authors mention, GNPTAB was already among the hits found in an earlier study using VSV-Ebola. Also, GNPTAB is known to be required for the function of Cathepsins which are well-know host factors required for Ebola virus infection. Therefore, this study mostly confirms a function for GNPTAB that was already expected. It would have been more interesting if this screen using authentic Ebola virus would have pointed out entirely new host factors that were needed for infection. However, due to the limited sensitivity of the screen we can also not conclude that VSV-Eb and authentic Ebola virus have the exact same host factor requirements for entry.

Together, I think that the experiments are solid and the conclusions are relevant and should be published. However, I am not convinced that the work should be published in Nature Communications.

We thank Reviewer 2 for these comments. Both reviewer 2 and reviewer 3 suggested a similar experiment, using a glycoprotein that does not require CatB and L for entry, and seeing if this can then enter GNPTAB-knockout cells. Reviewer 2 suggested using particles precleaved by CatB and L, whereas Reviewer 3 suggested using Reston virus (RESTV) glycoprotein. For simplicity, we elected to use the latter.

Since both of these reviewers suggested this similar experiment, the text of our draft may need clarifying on this point: while both the precleaved virus and RESTV are less dependent upon Cat B and L than other glycoproteins, they still require other cysteine protease(s) for entry. The evidence for this is that entry mediated by precleaved EBOV-GP is inhibited by E-64 (Shornberg et al, 2006, PMID 16571833; Kaletsky et al, 2007, PMID 17928356; Wong et al, 2010, PMID 19846533; Misasi et al, 2012, PMID 22238307). Similarly, while RESTV GP mediates entry into CatB^{-/-} CatL^{-/-} cells, RESTV entry is also inhibited by E-64 (Misasi et al, 2012).

The data is consistent with a multi-step model for EBOV glycoprotein, where initial cleavages by CatB and L expose the NPC1-binding domain, but additional E64-sensitive steps (possibly mediated by another lysosomal cathepsin) are required for fusion to occur. The RESTV glycoprotein may not require cleavage by CatB and L, but still requires E64-sensitive processing to occur. If a lysosomal protein, transported there through the action of GNPTAB, mediates these additional steps, one might expect that the entry of precleaved virus, and the growth of RESTV, in GNPTAB-deficient cells would be impaired.

To test this, we performed a time-course of RESTV growth in parental HAP1, GNPTAB⁻ and NPC1⁻ cells, and found that virus yields were up to ~200-fold lower in the GNPTAB-knockout cells than in parental HAP1s. Thus, the data is indeed consistent with another lysosomal protease being responsible for RESTV glycoprotein cleavage. We have included a graph of this experiment as Fig. S2g in our revised manuscript.

We have revised the text of our discussion to try to emphasize that the processing of the filovirus glycoproteins is a complex, multi-step process, relying on multiple lysosomal cysteine proteases:

“Lysosomal cysteine proteases are required to activate GP from all filoviruses, but different viruses appear to use different cathepsins. For example, the entry mediated by EBOV GP was strongly dependent upon CatB, whereas MARV GP was more dependent upon CatL⁴⁴. RESTV GP was not sensitive to the loss of CatB and CatL, but was inhibited by the broad cysteine protease inhibitor E-64, suggesting that an a different cysteine protease is required for RESTV infection⁴⁴. We found that RESTV infection of GNPTAB⁻ cells was impaired (Fig. S2g), consistent with RESTV GP activation being dependent upon a lysosomal protease, transported there through the action of GNPTAB. Additional cysteine proteases are required even for viruses that use CatB, as viruses bearing precleaved glycoproteins remain sensitive to inhibition by E-64^{44, 45, 46, 47}. This is consistent with a multi-step model where initial cleavages by CatB and L expose the NPC1-binding domain, but an additional E64-sensitive step(s), mediated by another lysosomal protease, is required for fusion to occur.”

We have reanalyzed our data and now provide the full details of the screening results (Supplementary files 1 and 2, selected hits in our revised Table 1). These do include some host factors that have not been identified previously, such as *SPNS1* which is a putative transporter protein implicated in lysosome function, and the *SLC30A1* gene which encodes the zinc-transporter ZNT-1. Further work will be required to confirm the role of these genes in EBOV infection, but our preliminary data suggest that *SPNS1* is required for EBOV infection of Huh7 cells, but is dispensable for infection of HAP1 cells. Some of the novel identified genes do make

mechanistic sense; *UVRAG*, for example, is known to be required for influenza A and VSV entry (Pirooz et al, 2014; PMID #24550300).

Finally, several of the genes identified in our screen, including *NPC1*, *GNPTAB* and HOPS-complex subunits, having originally been reported by Carette and colleagues using a pseudotype system, indicate that at least some host factors are shared between authentic and pseudotyped virus.

Reviewer #3 (Remarks to the Author):

In the manuscript “A genome-wide CRISPR screen identifies N-acetylglucosamine-1-2 phosphate transferase as a potential antiviral target for Ebola virus” by Flint et al. the authors identify N-acetylglucosamine-1-2 phosphate transferase (*GNPTAB*) as an essential host factor for the Ebola virus (EBOV) life cycle, and show that inhibiting the processing of this factor results in inhibition of EBOV entry and infection. The manuscript is very well written, and the experiments are straightforward and adequately address the scientific questions the authors pose. From a scientific point of view, there is little to be criticized, with exception of a few points as listed below.

The major issue with this study with respect to publication in Nature Communications is that of impact, importance to a wider field, and novelty. As the authors mention themselves, *GNPTAB* has been identified in a previous genome-wide haploid genetic screen in human cells (Carette et al., 2011, PMID 21866103). Further (and not mentioned by the authors), there are a number of other genome-wide screens for filovirus host-factors, although this study is to the best of my knowledge the first using CRISPR technology. Filone et al. reported in 2015 a genome-wide shRNA-based screen using infectious EBOV, although they did not identify *GNPTAB* (Filone et al., 2015, PMID 25646658). Similarly, the entry of a close relative of EBOV, Marburg virus (*MARV*), was studied using a genome-wide siRNA screen together with *MARV* GP-pseudotype viruses (Cheng et al., 2015, PMID 26596270). Again, *GNPTAB* was not reported as an identified factor; however, the authors of that study did not disclose the complete data set, but just a chosen subset of identified factors. Finally, Martin et al. recently reported a genome-wide siRNA screen for host factors important for the life cycle of EBOV, although in their screen they only assessed a role in genome replication and transcription, and not in entry (Martin et al., 2018, PMID 30081931).

In addition, to me it is not clear whether the (re-)identification of *GNPTAB* will be of interest to a wider field of virologists outside of the filovirus community. Only very limited data is presented that this factor plays a role for other viruses, although the authors discuss this possibility. Also, as a target for treatment *GNPTAB* it is certainly interesting; however, based on the presented data it is not clear whether such an approach in the end will be feasible. Therefore, while the final decision whether this manuscript meets the criteria of publication in Nature Communication has to be made on the editorial level, I personally am doubtful that this is the case.

Individual points of criticism that should be addressed prior to publication are:

1) Data availability: Since the authors performed a genome-wide screen, they should provide all the results of this screen. It is hard to believe that they identified only 4 host factors in their screen! If this is indeed the

case, they should discuss why their analysis yielded only so few hits, and disclose the primary data in order to allow others to mine this dataset.

As suggested, we have now provided all the primary data from our screen, both at the sgRNA level and at the gene level, which are now Supplementary files 1 and 2, respectively.

2) line 52: The authors' use of nomenclature is wrong. This sentence should read: "Ebolaviruses, such as Ebola virus (EBOV) and Reston virus (RESTV), and marburgviruses, such as Marburg virus (MARV), have a single-stranded ..."

We apologize for the incorrect nomenclature, and as suggested, have changed the text to read "Ebolaviruses, such as Ebola virus (EBOV) and Reston virus (RESTV), and marburgviruses, such as Marburg virus (MARV), have a single-stranded, negative-sense RNA genomes and are classified in the family *Filoviridae*."

3) The authors mention that RESTV apparently does not use Cathepsin B or L for its entry. Is this virus then still dependent on GNPTAB? The authors could (and should) easily check this in their pseudotype assay.

This experiment is essentially the same as that suggested by reviewer 2, who suggested using precleaved EBOV glycoproteins – please see our response to that comment for more details. Briefly, while RESTV GP-mediated entry appears to be independent of CatB and L, it is also sensitive to E-64 (Misasi et al, 2012), suggesting that a different lysosomal protease is responsible for RESTV GP activation. Consistent with this, we found that RESTV growth was impaired in GNPTAB-knockout cells (Fig. S2g). We have modified our discussion to expand and clarify on this point.

4) The authors should mention other (genome-wide) screens for filovirus host factors, and discuss why these screens did not pick up GNPTAB.

We now cite other published screens for filovirus host factors and compare hits from each screen, and our own, in our new Table 2. We also discuss reasons for differences between the screens; both of the screens that used a knockout method (retroviral gene-trapping and CRISPR-Cas9) found GNPTAB, whereas those screens that used knockdown through RNAi, did not. We now speculate in our discussion:

"Genome-wide screens can frequently yield discordant results^{39, 40}, with different techniques, cell-lines and challenge methods likely being sources of discrepancies. For example, retroviral gene-trapping and CRISPR-Cas9 methods are capable of inducing gene knockouts, whereas si- and shRNA use RNA interference (RNAi) to degrade transcripts. For a protein with a long half-life, or a very abundant transcript, a knockout screening method may be more effective than a knockdown. This may be why *GNPTAB* was a hit in two screens that used gene knockout, but not in those using RNAi (Table 2)."

REVIEWERS' COMMENTS:

Reviewer #1 (Remarks to the Author):

The authors have responded well and alleviated the minor concerns of this reviewer. I found the results of great interest to a broad research community.

Reviewer #3 (Remarks to the Author):

In the revised version of the manuscript “A genome-wide CRISPR screen identifies N-acetylglucosamine-1-2 phosphate transferase as a potential antiviral target for Ebola virus” the authors adequately have addressed all concerns raised by me.

Particularly, they now provide the complete data set for their screen, put their work in the context of previous siRNA screens, and have addressed the question of GNPTAB-dependence of Reston virus.

This has further improved a manuscript that already upon first submission was very good from a scientific perspective. Thus, aside from my previously raised concerns regarding impact, importance to a wider field, and novelty, there are in my view no other arguments speaking against a publication of this work.